# POLICY TREE NETWORK

## ABSTRACT

Decision-time planning policies with implicit dynamics models have been shown to work in discrete action spaces with Q learning. However, decision-time planning with implicit dynamics models in continuous action space has proven to be a difficult problem. Recent work in Reinforcement Learning has allowed for implicit model based approaches to be extended to Policy Gradient methods. In this work we propose Policy Tree Network (PTN). Policy Tree Network lies at the intersection of Model-Based Reinforcement Learning and Model-Free Reinforcement Learning. Policy Tree Network is a novel approach which, for the first time, demonstrates how to leverage an implicit model to perform decision-time planning with Policy Gradient methods in continuous action spaces. This work is empirically justified on 8 standard MuJoCo environments so that it can easily be compared with similar work done in this area. Additionally, we offer a lower bound on the worst case change in the mean of the policy when tree planning is used and theoretically justify our design choices.

## 1   INTRODUCTION

Reinforcement Learning (RL), the study of learning what to do. Learning what to do in a given situation so as to maximize reward signals. Generally speaking, Reinforcement Learning problems are approached from either a Model-Free or Model-Based perspective. A Model-Free approach builds a policy through interacting with the environment and uses the acquired experience to improve the policy. A Model-Based approach builds a dynamics model of the environment and uses this improve a policy in a number of ways. Such as planning, exploration, and even training on imagined data (Sutton, 1990; Ha & Schmidhuber, 2018). Model-Free Reinforcement Learning generally offers superior final performance. However, in exchange for strong final performance a large number of samples are required. Model-Based Reinforcement Learning offers better sample complexity but comes at the cost of a weaker final policy. In light of this, recent research (Wellmer & Kwok, 2019; Silver et al., 2016; Farquhar et al., 2017; Oh et al., 2017) has been focused on mixing both Model-Free and Model-Based Reinforcement Learning.

Model-Based Reinforcement Learning can be broken down further into two categories (Wellmer & Kwok, 2019). The first is explicit dynamics models. Explicit dynamics models are when the observations are directly being reconstructed. The second is implicit dynamics models. Implicit dynamics models are when the dynamics model is learned indirectly. Some examples of this could be through predicting future rewards, values, policies or other auxiliary signals.

The work we present here, Policy Tree Network, lies at the intersection of Model-Free and Model-Based Reinforcement Learning. Our work builds off the work done by Wellmer & Kwok (2019), where they showed how to build an implicit dynamics model for Policy Gradient methods in continuous action space. Policy Tree Network takes this a few steps further by showing how to leverage the implicit dynamics model for decision-time planning.

Our empirical results are on eight MuJoCo (Todorov et al., 2012) environments. In our experiments we validate the decision-time planning scheme that we introduce. Lastly we show Policy Tree Network outperforms the model-free baseline (Schulman et al., 2017) and the mixed model-free model-based baseline (Wellmer & Kwok, 2019).

## 2 RELATED WORKS

**Value Prediction Network** (VPN) (Oh et al., 2017) offers an approach to building and leveraging an implicit dynamics model in a discrete action space. The implicit dynamics model is constructed with overshooting objectives which predict future rewards and value estimates. At behavior time, a Q tree is expanded and actions are selected according to $\epsilon$-greedy of the backed-up Q estimates. Decision-time planning is trivial in this case since the action space is discrete. When the action space is discrete it's possible to try all possibilities.

**TreeQN and ATreeC** offer a policy gradient and Q-learning approach building implcit dynamics models in discrete action spaces. TreeQN (Farquhar et al., 2017) is similar to VPN except the authors claim that VPN has an issue since the policy being trained is different than the policy actually being used. In light of this, TreeQN directly optimizes the backed-up Q estimates as opposed to indirectly (as is done in VPN). Additionally, TreeQN uses overshooting reward estimates as an auxiliary objective.

ATreeC (Farquhar et al., 2017) takes a similar approach to TreeQN except instead of a Q-learning approach ATreeC uses a policy gradient approach. We note that even though ATreeC uses a policy gradient approach it still does not work with continuous action spaces. Crucially, ATreeC optimizes the backed-up "Q" estimates to function as logits for a multinomial distribution. Because of this, the backed-up "Q" values in this case can be thought of as pseudo Q values.

**Policy Prediction Network** (PPN) (Wellmer & Kwok, 2019) is an approach to building implicit dynamics models with policy gradient methods. This was the first work to show how to build an implicit dynamics model in continuous action spaces. This was done by introducing overshooting objectives and a clipping scheme designed for overshooting objectives. The training algorithm is located in Section A.2.

$$\mathcal{L}_t = \alpha_v \mathcal{L}_t^{d,v} + \sum_{i=0}^{d-1} (\mathcal{L}_t^{i,\pi} + \alpha_v \mathcal{L}_t^{i,v} + \alpha_r \mathcal{L}_t^{i,r}) \tag{1}$$

$$\mathcal{L}_t^{i,\pi} = \frac{1}{2} \max(-\text{ratio}_t^i A_{t+i}^{\text{GAE}}, -\text{ratio}_{t,\text{clip}}^i A_{t+i}^{\text{GAE}}) - \alpha_h H_t^i \tag{2}$$

$$\mathcal{L}_t^{i,v} = \frac{1}{2} \max((\hat{v}_{t,\theta}^i - R_{t+i})^2, (\hat{v}_{t,\text{clip}}^i - R_{t+i})^2) \tag{3}$$

$$\mathcal{L}_t^{i,r} = \frac{1}{2} \max((\hat{r}_{t,\theta}^i - r_{t+i})^2, (\hat{r}_{t,\text{clip}}^i - r_{t+i})^2) \tag{4}$$

Where $\mathcal{L}_t^{i,\pi}$, $\mathcal{L}_t^{i,v}$, and $\mathcal{L}_t^{i,r}$ are the policy, value, and reward losses grounded at time $t$ predicting $i$ steps into the future. Additionally, subscript $\theta$ refers to estimates from the latest model parameters, $H$ is policy entropy, $A_{t+i}^{\text{GAE}}$ is the generalized advantage estimate (Schulman et al., 2016), $r$ is a reward target, $\hat{r}$ is a predicted reward, $\hat{v}$ is a predicted value, and $R$ is a boot-strapped n-step return target. The $\alpha$ coefficients are hyper-parameters used to trade off importance of objectives. The clipped estimates are used to stay near estimates from old parameters($\theta$') and are further defined in Section A.3 along with the definition of the importance sampling ratio.

The short coming of PPN is that it did not lend itself well to leveraging the implicit dynamics model to perform decision-time planning and thus used the model-free policy to sample actions. Additionally, it does not make use of the $Q$ function, a separate measure of how "good" an action is at a given state, as discussed in Section 3.2.

## 3 POLICY TREE NETWORK

Policy Tree Network uses a combination of model-free and model-based techniques. Policy Tree Network builds off the PPN (Wellmer & Kwok, 2019) approach to learning an implicit-model. Actions during behavior time are chosen by a model-free policy. Learning is done with a model-based approach that follows the behavior policy's rollout trajectory. However, the test policy follows a model-based approach. An implicit transition model is embedded into the architecture. Through overshooting objectives, a dynamics model is learned and the collection of forward predictions offer the benefit of additional signal in the gradient updates.

Our novel contribution in this works is a decision-time planning algorithm that allows us to leverage the implicit dynamics model. The decision-time planning algorithm is based on building a tree of possible future values, actions, states, and rewards and then performing a tree backup algorithm we introduce. In previous works it was not possible to directly leverage the implicit model for decision time planning with policy gradient methods in continuous action space. Our empirical results in Section 4 demonstrate the advantage of PTN over the model-free baseline (PPO) and the mixed model-free & model-based baseline (PPN).

## 3.1 LEARNING

Policy Tree Network is trained over a collection of overshooting objectives. We follow an identical clipping approach (Section A.3) to the one found in PPN. Targets used for training are computed exactly the same way as in PPN. More explicitly, no aspect of training depends on the decision-time planning algorithm described later in Section 3.2. While this might be desirable, we describe why it is difficult in A.1. We additionally introduce a term $\beta$ Hafner et al. (2018) used to trade off importance of short and long-term predictions. Lastly, for simplicity we drop the extra value loss term found in PPN (Equation 1), leaving us with the following: $\mathcal{L}_t = \sum_{i=0}^{d-1} \beta_i (\mathcal{L}_t^{i,\pi} + \alpha_v \mathcal{L}_t^{i,v} + \alpha_r \mathcal{L}_t^{i,r})$. Where $\mathcal{L}_t^{i,\pi}$, $\mathcal{L}_t^{i,v}$, and $\mathcal{L}_t^{i,r}$ (equations 2, 3, 4) are the policy, value, and reward losses grounded at time $t$ predicting $i$ steps into the future.

The implicit dynamics model is jointly learned from the objectives in $L_t$ whenever $i > 0$. When $i = 0$ estimates are not required to pass through the transition network. In Section A.2 we show the training algorithms for PPO (Schulman et al., 2017), PPN (Wellmer & Kwok, 2019), and PTN which are quite similar.

PPN (Wellmer & Kwok, 2019) was shown to have returns drastically differ between training depth values. The point of $\beta$ is to stabilize returns over different choices of training depth (Section A.6).

## 3.2 DECISION-TIME PLANNING

Including branching in PTN is appealing because it would allow for decision-time planning. Furthermore, this *appears* to be a simple task since a transition model is already being learned. However, because the decision-time planning policy is difficult to directly measure with a PDF (Section A.1), we resort to performing decision-time planning exclusively at evaluation time.

Traditionally, at every state, the next action in a state is either determined by a policy, $\pi$, learned usually based on a policy gradient method, or an action value function $Q$ that induces a policy that takes actions with the largest $Q$ value. However, in PTN, we have access to independently learned policy and $Q$ functions.[1] There are three ways of using $\pi$ and $Q$ to create the final policy, $\pi_F$.

1. setting $\pi_F = \pi$ as done in PPN, (Wellmer & Kwok, 2019), which ignores the $Q$ function.
2. setting $\pi_F = $ argmax $Q$, called $Q$-backup which ignores $\pi$[2].
3. setting $\pi_F = f(\pi, Q)$, called $\pi$-$Q$-backup, for some function $f$.

As (1) was described in PPN we will further dig into the details of (2) in Section 3.2.1 and (3) in Section 3.2.2. Methods discussed in this section do not impact the learning procedure discussed in Section 3.1 and are only used for decision-time planning at evaluation time.

We use a tree planning method which expands a tree up to a depth $d$ expanding $b$ branches at each depth and collecting reward, policy, and value predictions along the way. Tree expansion is done by recursively calling the core module ($f^{\text{core}}$) with $b$ action samples and the abstract state. The action being passed to the core module could be sampled from a uniform distribution or the policy ($\pi(a|\hat{s}_{t,\theta}^{i,j})$), where $\hat{s}_{t,\theta}^{i,j}$ represents the predicted abstract-state $i$ steps into the future with branch $j$ according to parameters $\theta$. We show that if the correlation between $\pi$ and $Q$ is positive, sampling $\pi$ will be more suitable and Section 4.2 shows that they are in fact positively correlated in practice. Candidate actions represent branches in the tree. Then the predicted estimates are backed-up with a TD-$\lambda$ scheme (Sutton, 1988; Sutton & Barto, 2018; Farquhar et al., 2017) to assign a Q value to each branch as described in Algorithm 1.

---

[1]by independent we mean $\pi$ is not just a policy induced by selecting max of Q.
[2]when $b = \infty$

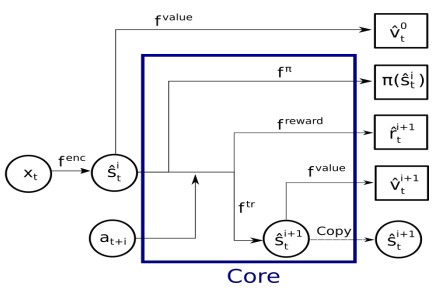

Figure 1: Predict policies, rewards, abstract states, and the value of the abstract states (Wellmer & Kwok, 2019).

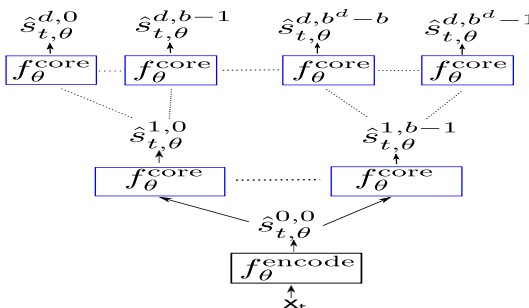

Figure 2: PTN Expansion, $a, r, v$ outputs from $f^{core}$ are omitted for simplicity

When performing decision-time planning, the action associated with the max backed-up Q-value is taken. This policy is referred to as $\pi_{\theta', F}$, where $\theta'$ denotes which parameters were used in the tree.

### 3.2.1 Q BACKUP

Given an abstract state $s = f_\theta^{enc}(x)$ and an action $a$, the backed-up Q-value calculated from $d$-step planning is defined as:

$$Q^i(s,a) = r + \gamma V^i(s') \quad (5) \quad V^i(s) = \begin{cases} v & \text{if } i = d \\ (1-\lambda)v + \lambda \max_a Q^{i+1}(s,a) & \text{if } i < d \end{cases} \quad (6)$$

where $s' = f_\theta^{tr}(s,a), v = f_\theta^v(s)$, and $r = f_\theta^r(s,a)$.

In Q backup, the final policy, $\pi_F$, aims at selecting an action that maximizes the action value function, $Q^0$. However, $Q^0$ is a combination of neural networks and finding $\arg\max_a Q^0$ is a difficult problem. We use our tree expansion to achieve this. With tree expansion, at each depth $i$, we sample $b$ actions based on $\pi$ and choose the one with maximum $Q^{i+1}$ to estimate $\arg\max_a Q^{i+1}$ in equation 6.

Intuitively, if $\pi$ and $Q$ are positively correlated, sampling from $\pi$ allows us to find actions with higher $Q$ compared with sampling from a uniform distribution, because, for a state $s$ and an action $a$, larger $\pi(a|s)$ value would suggest larger $Q(s,a)$. This is the subject of the following lemma. Note that, for a given state $s$, the covariance between $\pi(X|s)$ and $Q(X,s)$ over a bounded action space is defined as $cov_U(\pi(X|s), Q(X,s))$ where $X$ is a random variable with distribution $U$, the uniform distribution over the action space. $\pi$ and $Q$ are positively correlated if $cov_U(\pi(X|s), Q(X,s))$ is positive. Proof of the lemma can be found in the appendix.

**Lemma 1.** $E_{X\sim\pi}[Q(X,s)] > E_{X\sim U}[Q(X,s)]$ if $cov_U(\pi(X|s), Q(X,s)) > 0$.

Section 4.2 shows that the correlation between $\pi$ and $Q^0$ is in fact positive in practice. Therefore, sampling actions according to $\pi$ is justified.

Since sampling is done based on $\pi$, our final policy $\pi_F$ now depends on $\pi$ as well as $Q^0$. Next we investigate the relationship between $\pi_F$, $\pi$ and $Q^0$. Consider the abstract state $s = \hat{s}_{t,\theta}^{0,0}$. Recall that $Q^0(a,s)$ is an action value function calculated based on the tree expansion and backup. $Q^0$ uses sampling, is probabilistic, and is a function of $f_\theta^{core}$, $\lambda$, $b$ and $d$. Action selection is done by sampling $X_i \sim \pi$ for $1 \le i \le b$ and selecting $\arg\max_i Q^0(X_i, s)$. Thus, for an action $a$, cumulative density function (cdf) of $\pi_F(a|s)$ is given by $\int_{-\infty}^{a} \pi_F(z|s)dz = Pr(\arg\max_i Q^0(X_i, s) \le a)$.

Intuitively, the branching factor $b$ can be thought of as interpolating how much *confidence* we have in $\pi$ and reward, value and transition networks (which make up $Q^0$): low $b$ signifies confidence in $\pi$ and larger $b$ signifies more confidence in reward, value and transition networks. As $b$ increases $\pi_F$ becomes less dependant on $\pi$. When $b$ goes to infinity, $\pi_F$ only depends on reward, value and transition networks and not $\pi$. To investigate the maximum possible difference between mean of $\pi$ and $\pi_F$ over all $Q^0$ functions we provide the next theorem. The theorem is provided in a one dimensional action space for better illustration, but the argument can be extended to higher dimensions. Proof of the theorem can be found in the appendix.

**Theorem 1.** *Let $\mu_F$ be the mean of actions selected based on $\pi_F$ and $\mu$, $\sigma$ the mean and standard deviation of actions selected based on $\pi$. In the worst-case over all possible $Q^0$ functions, $|\mu - \mu_F| \ge$*

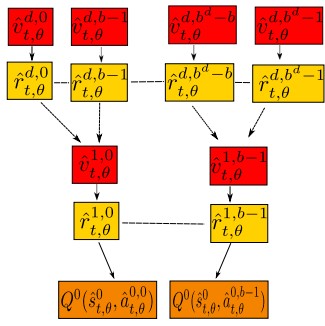

Figure 3: PTN Backup

**Algorithm 1** $\pi\_Q\_\text{expand}(s, i)$

initialize $\hat{a}, r, s, v', Q$
**for** j in b **do**
    $\hat{a}[j] \sim \pi(s)$
    $\hat{r}[j] = f_\theta^r(s, \hat{a}[j])$
    $\hat{s}'[j] = f_\theta^{tr}(s, \hat{a}[j])$
    $\hat{v}'[j] = f_\theta^{va}(\hat{s}'[j])$
    **if** $i + 1 == d$ **then**
        $Q[j] = \sqrt{\left(\hat{r}[j] + \gamma \hat{v}'[j]\right)\pi(a = \hat{a}[j]|s)}$
    **else**
        $Q\_\text{tmp} = \pi\_Q\_\text{expand}(\hat{s}'[j], i + 1)$
        $Q[j] = \sqrt{\left(\hat{r}[j] + \gamma((1 - \lambda)\hat{v}'[j] + \lambda \max_{\hat{a}} Q\_\text{tmp})\right)\pi(a = \hat{a}[j]|s)}$
**return** Q

Figure 4: pseudo-code for $\pi$-Q expansion and backup algorithm. Where the $Q$ returned is a $b$ dimensional vector.

$b|\sigma \int_{-\infty}^{\infty} z\phi(z)\left(\frac{erfc(z)}{2}\left(\frac{z}{\sqrt{2}}\right)\right)^{b-1}dz|$, *where erfc is the complementary error function and $\phi$ is the p.d.f of standard normal distribution. Setting $b = 2$, we get the looser bound $|\mu - \mu_F| \geq \frac{\sigma}{\sqrt{\pi}}$ in the worst-case.*

The bound is a lower bound on the worst-case and it shows that $\mu_F$ *can* move at least as far as $\frac{\sigma}{\sqrt{\pi}}$ from $\mu$. In other words, there exist $Q^0$ functions for which the difference between mean of $\pi_F$ and $\pi$ is at least $\frac{\sigma}{\sqrt{\pi}}$. Observe that the bound decreases as $\sigma$ decreases, which intuitively makes sense since with lower $\sigma$, it takes more number of samples to obtain an action further away from $\mu$.

The bound above sheds a light on one of the downsides of this approach. The final policy $\pi_F$ *can* become significantly different from $\pi$, depending on $Q^0$ and $\sigma$, even when $b = 2$ and as $b$ grows it becomes only reliant on $Q^0$ (Section A.5 shows the numerical value of the bound for several $b$ values). Although Section 4.2 shows empirically that $\pi$ and $Q^0$ are *on average* correlated, there can be states where they are not. In such a scenario, Q backup may rely too heavily on $Q^0$.

We previously mentioned that $b$ can be seen as a confidence parameter, adjusting how much we rely on $\pi$ and the networks that make up $Q^0$. Nevertheless, it does not remedy the above mentioned problem. This is for two reasons, one is that $b$ only takes discrete values. We mentioned above that $b = 2$ may shift the policy by too much, but we cannot reduce $b$ any more without exclusively relying on $\pi$. Secondly, choosing a $b$ for a given confidence level is a difficult problem, even if a discrete confidence parameter is sufficient. This is because $b$ has a complicated relationship with $\pi_F$, $\pi$ and $Q$ involving integrals. Thus, it is not clear that, for instance, if we have equal confidence in $Q$ and $\pi$, what the value of $b$ should be.

### 3.2.2 $\pi$-Q BACKUP

Motivated by the issues discussed in section 3.2.1, we introduce a $\pi$-Q backup. Given a policy $\pi$ and a Q-function $Q$, we want our final policy $\pi_F$ to depend on $Q$ and $\pi$ based on how much confidence we have in each. If we have equal confidence in both $Q$ and $\pi$, then we would like $\pi_F$ to be a function depending on $\pi$ and $Q$ *equally*. One way to achieve this is using geometric mean of $\pi$ and $Q$ as an indicator of how good an action is [3]. That is, we would like to select an action such that $\sqrt{\pi \times Q}$ is maximized (note that if we have reasons to believe $\pi$ or $Q$ provide better estimates, we can take the weighted geometric mean).

The backup procedure is defined as mentioned in Section 3.2.1 , however, given an abstract state $s = f_\theta^{enc}(x)$ and an action $a$, the backed-up Q-value calculated from $d$-step planning is defined as:

$$Q^i(s, a) = \sqrt{(r + \gamma V^i(s'))\pi(a|s)} \quad (7) \quad V^i(s) = \begin{cases} v & \text{if } i = d \\ (1 - \lambda)v + \lambda \max_a Q^{i+1}(s, a) & \text{if } i < d \end{cases} \quad (8)$$

Note that neither $Q^i$ or $V^i$ in Equations (5, 6, 7, 8) are directly being optimized, they are only used for decision-time planning. Notice how estimates are scaled by likelihood of the action being sampled from $\pi$. This removes the previously mentioned issue of selected actions no longer depending on $\pi$. It's interesting to note that in this case when $b = \infty$ the policy is deterministic.

---

[3] When a negative or zero valued Q exists in the tree backup, modify the geometric mean by adding $|\min(Q)| + \delta$ to the Q estimates. $\delta$ is a positive small non-zero term which can be optimized Habib (2012)

This does not manifest itself as an issue because the decision-time planning policy isn't used as the behavior policy so we are less concerned about exploration.

Note that we still need to estimate $\max_a Q^{i+1}$ in equation 8. We take a similar approach to Q-backup mentioned in Section 3.2.1, since $\pi$ and $\sqrt{\pi Q}$ are expected to be correlated. Furthermore, regarding the use of geometric mean, we note that it is more meaningful than the arithmetic mean in this scenario as the values may be of different scales and that it preserves proportional change as opposed to absolute change.

In Section 4.3 we will empirically explore both approaches to backup. The $\pi$-$Q$ backup is shown in Algorithm 4. Though we note that if the scaling of $Q$ estimates by $\pi$ and the square root are dropped it would reduce to the max $Q$-backup algorithm described in Section 3.2.1.

## 4 EXPERIMENTS

Our experiments seek to answer the following questions: (1) Is correlation between $\pi$ and $Q$ positive? (2) What style of backup performs better: a standard Q-value backup or a policy weighted Q-value backup and how does branching effect the returns of the decision-time planning policy? (3) Does PTN outperform the baselines?

### 4.1 EXPERIMENTAL SETUP

Preprocessing in our experiments was done similarly to that of PPO (Schulman et al., 2017) and identical to the preprocessing in PPN (Wellmer & Kwok, 2019). All models, PPO2, PPN, and PTN were implemented in Pytorch (Paszke et al., 2017). The parameters ($\theta$) are updated with the Adam optimizer (Kingma & Ba, 2014). All the experiments are run for 1 million time steps unless otherwise noted.

Our PPO2 implementation uses the same hyperparameters as the baselines implementation (Dhariwal et al., 2017): 3 fully connected layers with 128 hidden units and tanh activations for the policy. 3 fully connected layers 128 hidden units and tanh activations for the critic. The largest difference in our PPO2 implementation is that we do not perform orthogonal initialization.

Our PTN implementation uses similar hyperparameters (identical to PPN): 2 fully connected layers with 128 hidden units and tanh activations for the embedding. 2 fully connected residual layers with 128 hidden units, tanh activations, and unit length projections of the abstract-state (Farquhar et al., 2017) for the transition module. 1 fully connected layer with 128 hidden units for the policy mean. 1 fully connected layer with 128 hidden units for the value. 1 fully connected layer with 128 hidden units for the reward. In practice we use Huber losses instead of L2 losses, as was done in related implicit model based works (Oh et al., 2017).

We set $\beta_0 = d$ and $\beta_{>0} = 1$. Values used for branching and depth are explicitly stated in corresponding experiments. We test on the 8 standard OpenAI MuJoCo environments (Todorov et al., 2012; Brockman et al., 2016): Hopper, Walker2d, Swimmer, HalfCheetah, InvertedPendulum, InvertedDoublePendulum, Humanoid, and Ant.

### 4.2 CORRELATION

|  | sample correlation mean | sample correlation variance |
|---|---|---|
| Hopper-v2 | 0.937 | 0.018 |
| Walker2d-v2 | 0.971 | 0.017 |
| Swimmer-v2 | 0.945 | 0.016 |
| HalfCheetah-v2 | 0.968 | 0.018 |
| InvertedPendulum-v2 | 0.969 | 0.008 |
| InvertedDoublePendulum-v2 | 0.971 | 0.005 |
| Humanoid-v2 | 0.934 | 0.022 |
| Ant-v2 | 0.972 | 0.019 |

Table 1: Summary of correlation between $Q$ and $\pi$.

In this experiment we empirically measure the correlation between $\pi$ and $Q^0$(with $d = 1$). Positive correlation empirically justifies the use of lemma 1 that sampling from $\pi$ offers a better approach to

maximizing the backup objective (for both Q backup and $\pi$Q backup) as opposed to uniform sampling. To measure correlation we first train a PPN policy for 200 thousand time-steps on the MuJoCo environments. Next we collect trajectories over 10 thousand timesteps. Then at each observation we uniformly sample 100 actions and compute the corresponding $Q$-values and corresponding PDF points from $\pi$. This gives us 10 thousand estimates of correlation. From here we fit a normal distribution to the correlation results and report them in Table 3.

Positive correlation tells us that sampling according to $\pi$ will better maximize $\pi_F$ than a uniform sample over actions. This empirically justifies our approach to performing tree expansion.

## 4.3 BRANCHING & BACKUP

| | PTN $\pi$-$Q$ backup | | | | PTN $Q$ Backup | | | |
|---|---|---|---|---|---|---|---|---|
| | b=1 | b=2 | b=3 | b=5 | b=1 | b=2 | b=3 | b=5 |
| Hopper | 2296.8 ± 403.6 | 2784.4 ± 357.8 | 2981.3 ± 306.7 | 3083.5 ± 244.1 | 2155.9 ± 343.9 | 1763.0 ± 431.9 | 1400.8 ± 423.5 | 1103.4 ± 398.1 |
| Walker2d | 3442.7 ± 182.0 | 4080.3 ± 295.5 | 4189.1 ± 355.9 | 4253.8 ± 334.3 | 3381.5 ± 378.3 | 3539.7 ± 517.6 | 3160.7 ± 743.5 | 2722.8 ± 1033.0 |
| Swimmer | 102.1 ± 28.4 | 105.6 ± 29.2 | 106.6 ± 29.2 | 107.5 ± 29.2 | 89.5 ± 35.1 | 88.6 ± 34.4 | 84.5 ± 32.0 | 75.5 ± 30.9 |
| HalfCheetah | 4082.1 ± 659.9 | 4346.3 ± 649.1 | 4459.8 ± 638.8 | 4555.0 ± 643.6 | 3435.8 ± 742.4 | 3501.7 ± 748.2 | 3526.3 ± 734.5 | 3467.7 ± 741.1 |
| InvertedPendulum | 1000.0 ± 0.0 | 1000.0 ± 0.0 | 1000.0 ± 0.0 | 1000.0 ± 0.0 | 995.2 ± 9.6 | 998.7 ± 2.6 | 998.6 ± 2.8 | 998.2 ± 3.6 |
| InvertedDoublePendulum | 6301.4 ± 299.8 | 9347.6 ± 0.2 | 9353.2 ± 0.2 | 9356.9 ± 0.1 | 6286.6 ± 308.4 | 9244.3 ± 225.1 | 9159.7 ± 225.1 | 7929.4 ± 1498.2 |
| Ant | 1730.3 ± 225.7 | 2109.3 ± 329.7 | 2183.2 ± 324.8 | 2370.0 ± 371.3 | 1946.4 ± 266.1 | 2047.1 ± 202.7 | 2078.5 ± 197.5 | 2013.3 ± 279.1 |
| Humanoid | 747.8 ± 280.7 | 788.0 ± 355.6 | 817.8 ± 410.5 | 895.5 ± 566.3 | 933.1 ± 432.9 | 957.3 ± 449.2 | 919.2 ± 345.9 | 859.8 ± 272.9 |

Table 2: branching and backup experiment where $d = 2$ and $\lambda = 0.95$

In this section we explore the impact of using planning as the test policy and which backup scheme captures the highest returns. We consider a PTN trained with a depth equal to 2 and we evaluate both backup procedures found in Section 3.2.

As we can see in Table 2, nearly all environment suffer from an increased branching factor when using Q backup. None of the environments show a clear trend that increasing the branching benefits returns. In fact, both Hopper and Walker2d environments show the opposite, as $b$ increases returns decrease. Intuitively this makes sense, as $b$ increases to infinity the action taken no longer relies on the policy $\pi$ but instead relies entirely on the $Q$ backup from reward and value predictions.

When the backup scheme takes into account the likelihood of an action coming from the policy, it is no longer true that as $b$ increases to infinity the action selected by decision-time planning is independent from the policy. As we can see in Table 2, every environment benefits from increasing branching used in decision-time planning when the backup scheme is $\pi$-$Q$ backup discussed in Section 3.2.2. Comparing $Q$-backup and $\pi$-$Q$-backup in Table 2 it's obvious to see that the policy weighted Q backup offers much more robust returns.

Now we clearly see that decision-time planning is indeed helpful. Furthermore, we now know that a $\pi$-$Q$ backup offers a reliable approach. Going forward we will ask if it's possible to learn from this decision-time planning policy.

## 4.4 BASELINE COMPARISON

To test our model, we chose to benchmark against PPO2 and PPN on eight MuJoCo (**?**) environments. We include our $d = 2$ $b = 5$ model in our baseline comparison. However, we note that it is possible that other configurations perform better on some environments.

As can be seen in Figure 5, we find that PTN finds a better if not comparable policy in all of the environments. PTN has large performance gains over PPO's model-free approach, even though PTN's implicit planning module only looks a short distance into the future.

Environments with larger meaningful observation spaces, such as Humanoid, we suspect could benefit from a bigger latent space. In Humanoid, the latent space (128) is far smaller than the observation dimensions (376) and makes learning a useful implicit transition model infeasible. Given more compute we would explore increasing the latent space.

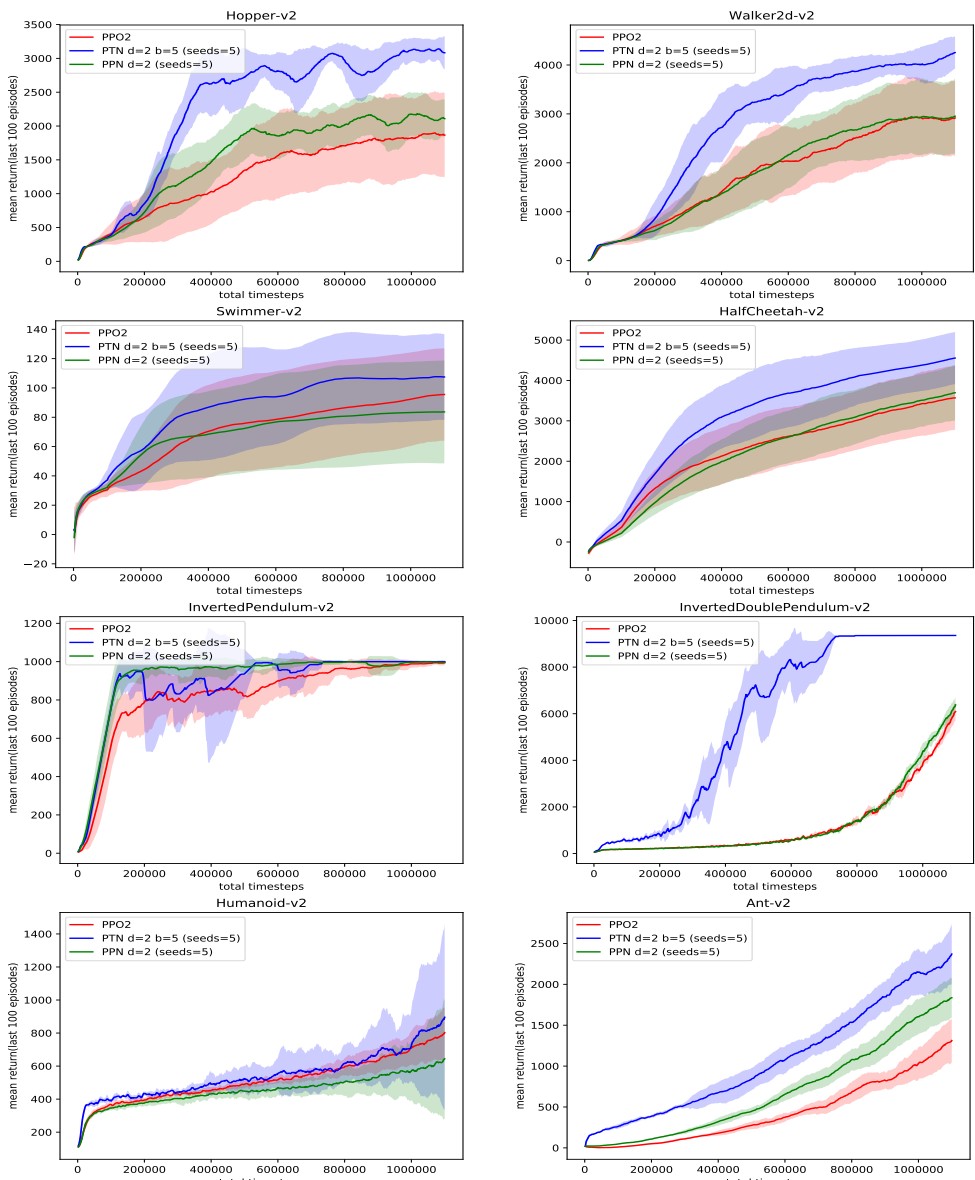

Figure 5: Results on 1 million step MuJoCo benchmark. Dark lines represent the mean return and the shaded region is a standard deviation above and below the mean.

## 5 CONCLUSION

In this work we present for the first time an approach to decision-time planning for implicit dynamics models in continuous action space. We provide a theoretical justifications for our design choices and show strong empirical results which are improvements over previous related work. In future work we would like to investigate distilling the tree policy into the model-free policy, find a better sampling procedure for expansion, and incorporate research from recurrent networks to avoid issues with vanishing and exploding gradients in the implicit dynamics model.

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

# A APPENDIX

## A.1 DECISION-TIME PLANNING AS THE BEHAVIOR POLICY

Including branching in Policy Tree Network's behavior policy is appealing because it would allow for directly optimizing the test policy we intend to use and *appears* to be simple since an implicit transition model is already learned. However, as we will soon see this is not the case.

A natural first thought is just to recursively sample $\pi$ $b$ times up to a depth $d$, perform a backup on the expanded tree, and then take the action associated with the maximal base branch. This is the approach described in Section 3.2.

However, this can not be used as the behavior policy because you are changing the distribution of how rollout actions are chosen in a way that is difficult to directly measure. The rollout policy $\pi_{\theta',F}$ is not equal to $\pi_{\theta'}$ used to do importance sampling in the policy gradient loss. When $b = 1$ then $\pi_{\theta',F}$ is the same as $\pi_{\theta'}$. However as soon as $b > 1$ we begin to rely on the reward and value network to help decide which actions to take.

The previously mentioned issues can be avoided if $\pi_{\theta',F}$ is not used as the behavior policy. Instead with decision-time planning we can only use $\pi_{\theta',F}$ as our test policy. Interestingly, there is no guarantee that this should work because the policy we are optimizing for ($\pi_\theta$) is not the same as the test policy. We note that a theoretical issue does arise when using $\pi_{\theta',F}$ as the test policy. The $f_{\theta'}^v$ network parameters are trained to predict the value for policy $\pi_{\theta'}$ not $\pi_{\theta',F}$. However, in practice this turns out not to be a large issue.

## A.2 PPN TRAINING ALGORITHM

---
**Algorithm 2** Policy Prediction Network(PPN) (Wellmer & Kwok, 2019)
---
Initialize parameters $\theta$
$\theta' = \theta$
**for** iteration=1, 2, . . . **do**
    Run policy $\pi_{\theta'}$ in environment for $n$ time steps
    Compute advantage estimates $A_1^{GAE}, \ldots, A_n^{GAE}$
    **for** epoch= 1, . . . , $K$ **do**
        Shuffle $n$ samples into mini-batches of size $M \leq n$
        **for** each mini-batch **do**
            $T$ is the set of samples selected for the mini-batch
            $\mathcal{L}_{mb} = \frac{1}{M} \sum_{t \in T} \mathcal{L}_t$
            Optimize $\mathcal{L}_{mb}$ w.r.t. $\theta$
    $\theta' = \theta$
---

PPO, PPN, and PTN use similar training algorithms, the main difference stems from how $\mathcal{L}_t$ is defined. Where in Algorithm 2, $n$ time steps are used to collect trajectories, $K$ is the number of epochs, $M$ is mini-batch size, and $T$ represents the randomly sampled time steps in a specific mini-batch.

## A.3 CLIPPING

$$
\begin{aligned}
\text{ratio}_t^i &= \frac{\pi_\theta(a = a_{t+i}|s = \hat{s}_{t,\theta}^i)}{\pi_{\theta'}(a = a_{t+i}|s = \hat{s}_{t+i,\theta'}^0)} \\
\text{ratio}_{t,\text{clip}}^i &= \text{clip}(\text{ratio}_t^i, 1 - \epsilon, 1 + \epsilon), \\
\hat{v}_{t,\text{clip}}^i &= \text{clip}(\hat{v}_{t,\theta}^i - v_{t+i,\theta'}^0, -\epsilon, \epsilon) + \hat{v}_{t+i,\theta'}^0, \\
\hat{r}_{t,\text{clip}}^i &= \text{clip}(\hat{r}_{t,\theta}^i - \hat{r}_{t+i,\theta'}^0, -\epsilon, \epsilon) + \hat{r}_{t+i,\theta'}^0.
\end{aligned}
$$

During training we follow the grounded clipping approach shown in PPN (Wellmer & Kwok, 2019). The clipped estimates are defined above. Where $\epsilon$ is a hyperparameter that defines the size of the clipping region. Clipping all the network heads turns out to be imperative to the learning process (Wellmer & Kwok, 2019).

## A.4 PROOFS

*Proof of Lemma 1.* Note that $cov_U(\pi(X|s), Q(X,s)) = E_U[Q(X,s)\pi(X|s)] - E_U[Q(X,s)]E_U[\pi(X|s)]$. Let $R$ be a bounded action space. By definition, the covariance equals $\frac{1}{area(R)}\int_{x\in R}(Q(x,s)\pi(x|s)) - E_U[Q(X,s)]\frac{1}{area(R)}\int_{x\in R}\pi(x|s)$, where $\int_{x\in R}\pi(x|s) = 1$ and $\int_{x\in R}(Q(x,s)\pi(x|s)) = E_\pi[Q(X,s)]$. The result follows because the covariance is positive.

$\square$

*Proof of Theorem 1.* We prove this by showing the existence of $Q^0$ functions for which the inequalities holds. We drop $s$ from our notation for convenience and write $Q(a,s)$ as $Q(a)$ and $\pi(a|s)$ as $\pi(a)$.

Let $X_F \sim \pi_F$, i.e., $X_F = \arg\max_i Q^0(X_i)$, where $X_i \sim \pi$. Consider a $Q^0(a)$ that is strictly decreasing as $a$ increases. Then $\int_{-\infty}^c \pi_F(x)dx = Pr(X_F \le c) = Pr(\exists i, X_i \le c) = 1 - Pr(\forall_i, X_i > c)$. Since $X_i$'s are i.i.d, $Pr(\forall_i, X_i > c) = Pr(X > c)^b = \left(\frac{\text{erfc}(\frac{c-\mu}{\sqrt{2\sigma^2}})}{2}\right)^b$ (setting $X = X_1$). Therefore, $\pi_F(x) = b\left(\frac{\text{erfc}(\frac{c-\mu}{\sqrt{2\sigma^2}})}{2}\right)^{b-1}\pi(c)$.

Then $\mu_F = b\int_{-\infty}^\infty c\left(\frac{\text{erfc}(\frac{c-\mu}{\sqrt{2\sigma^2}})}{2}\right)^{b-1}\pi(c)dc$. Substitute $c = \sigma z + \mu$, we have

$$\mu_F = b\sigma \int_{-\infty}^\infty z\left(\frac{\text{erfc}(\frac{z}{\sqrt{2}})}{2}\right)^{b-1}\phi(z)dz + b\mu \int_{-\infty}^\infty \left(\frac{\text{erfc}(\frac{z}{\sqrt{2}})}{2}\right)^{b-1}\phi(z)dz$$

where,

$$b\mu \int_{-\infty}^\infty \left(\frac{\text{erfc}(\frac{z}{\sqrt{2}})}{2}\right)^{b-1}\phi(z)dz = \mu\left(\frac{\text{erfc}(\frac{z}{\sqrt{2}})}{2}\right)^b\Big|_{-\infty}^\infty$$
$$= 0 - (-\mu)$$

and when $b = 2$,

$$(2)\sigma \int_{-\infty}^\infty z\left(\frac{\text{erfc}(\frac{z}{\sqrt{2}})}{2}\right)^{(2)-1}\phi(z)dz = 2\sigma\left(\frac{\text{erfc}(\frac{z}{\sqrt{2}})}{2}\right)\phi(z) - \frac{\text{erf}(z)}{4\sqrt{\pi}}\Big|_{-\infty}^\infty$$
$$= -\frac{\sigma}{\sqrt{\pi}}$$

Observe that $b\int_{-\infty}^\infty \left(\frac{\text{erfc}(\frac{z}{\sqrt{2}})}{2}\right)^{b-1}\phi(z)dz = 1$, which gives the first bound.

Setting $b = 2$, we have that $\mu_F - \mu = -\frac{\sigma}{\sqrt{\pi}} + \mu - \mu = -\frac{\sigma}{\sqrt{\pi}}$.

A similar argument but setting $Q^0$ to be a strictly increasing function in $a$ gives the inequality for when $\mu_F$ is larger than $\mu$.

$\square$

## A.5 NUMERICAL EVALUATION OF BOUND IN THEOREM 1

Fig. 6 shows the numerical values for the bound in Theorem 1 for several branching values. Note that the bound becomes larger than $1 \times \sigma$ when $b = 4$. This means that when $b = 4$, the difference between the mean of $\pi_F$ and $\pi$ can become larger than one standard deviation of $\pi$, which is a significant difference.

## A.6 LEARNING MODIFICATIONS

Notice that these returns are worse than PTN(with decision-time planning) shown in Figure 5. This shows performance benefits from the modifications shown in Section 3.1(ex: $\beta$). The point of

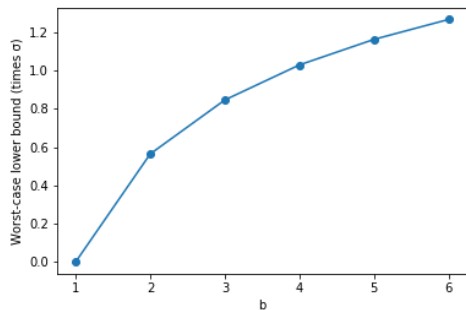

Figure 6: Numerical evaluation of bound in Theorem 1 for various branching values

| | train depth = 5 | | train depth = 10 | |
|---|---|---|---|---|
| | PTN | PPN | PTN | PPN |
| Hopper-v2 | $1944.0 \pm 258.5$ | $\mathbf{2191.6 \pm 183.3}$ | $1672.7 \pm 457.5$ | $\mathbf{1752.3 \pm 625.4}$ |
| Walker2d-v2 | $\mathbf{2936.9 \pm 893.2}$ | $2808.4 \pm 647.3$ | $\mathbf{3054.8 \pm 434.0}$ | $2565.9 \pm 327.9$ |
| Swimmer-v2 | $\mathbf{89.4 \pm 35.3}$ | $73.3 \pm 34.5$ | $\mathbf{80.3 \pm 35.7}$ | $57.5 \pm 30.3$ |
| HalfCheetah-v2 | $\mathbf{3439.8 \pm 698.6}$ | $3410.5 \pm 840.9$ | $\mathbf{3638.9 \pm 545.2}$ | $3632.8 \pm 819.0$ |
| InvertedPendulum-v2 | $971.7 \pm 56.5$ | $\mathbf{979.6 \pm 25.7}$ | $1000.0 \pm 0.0$ | $1000.0 \pm 0.0$ |
| InvertedDoublePendulum-v2 | $\mathbf{4332.6 \pm 192.9}$ | $3344.5 \pm 234.4$ | $\mathbf{4360.2 \pm 145.1}$ | $3186.8 \pm 87.7$ |

Table 3: Returns from PTN(no $\pi_F$/decision-time planning) and PPN using only the model-free policies

$\beta$(found in PTN but not PPN) is to stabilize returns over different values of training depth. In PPN, Wellmer & Kwok (2019) showed that optimal depth is highly dependent on the environment and returns can drastically differ. While the modifications certainly do not entirely fix this, we find that it mitigates large differences in returns from different training depths.

