# OpenReview forum: "Policy Tree Network"
_ICLR.cc/2020/Conference — Reject_

### Official Review · AnonReviewer1 · 2019-10-23
**Official Blind Review #1**

**Rating:** 3

**Review:**

This paper presents Policy Tree Network, a novel approach to use an implicit dynamics model to perform decision-time planning in continuous action spaces. The experiments show that the proposed method performs better than the underlying model-free RL algorithm in standard MuJoCo environments.

The writing quality is low and I don't understand the proposed method, especially the backed-up Q-value. The notation is also confusing. It seems to me that pi(a|s) is the density of action a, so what does sqrt(value * density) mean? What if the value is negative?

Besides, I have some questions.
1. Figure 1 looks the same as Figure 1 in https://arxiv.org/pdf/1909.07373.pdf, but I don't find any reference in the paper. Could you please state the difference between two figures, or explain why you want to put this figure here without any description or reference?
2. How is PTN compared to model-based RL algorithms? The only baseline here is PPO, which is model-free. More importantly, note that the policy in PPO is stochastic, so how is PTN compared to the deterministic policy?
3. How is the proposed pi-Q-backup method compared to classical control method, e.g. MPC? Does the proposed planning algorithm work for model-free algorithms?
4. As this paper talks about planning with implicit dynamics models, how is the proposed method compared with explicit dynamics models?

Minor comments:
1. In Algorithm 1 Line 11, could you please check the brackets?
2. Page 4, "Thus, cumulative density function (cdf) of pi_F is given by ...": Could you please check the correctness of the equation?
3. What does "worst-case" in Theorem 1 mean?
4. How is correlation in Table 2 calculated?
5. In Algorithm 1, is the return value a scalar or a vector?
6. The paper states that "Intuitively, the branching factor b can be thought of as interpolating how much confidence we have in pi and Q0". One can have infinite b but sample a uniformly to optimize Q (and then pi_F becomes maxQ policy), so I don't think b can be simply characterized as the confidence.


**Experience Assessment:**

I have published one or two papers in this area.

**Review Assessment: Checking Correctness Of Derivations And Theory:**

I assessed the sensibility of the derivations and theory.

**Review Assessment: Checking Correctness Of Experiments:**

I assessed the sensibility of the experiments.

**Review Assessment: Thoroughness In Paper Reading:**

I read the paper at least twice and used my best judgement in assessing the paper.

---

> ### Author Response · Authors · 2019-11-13
> **Response to Reviewer #1**
>
> Thanks for taking the time to review and give feedback. We’ve reviewed your points and will respond in line below.
>
> > It seems to me that pi(a|s) is the density of action a, so what does sqrt(value * density) mean?
>
> Consider value and density functions as two different quality scores for an action. We use the geometric mean to provide an average quality score for the action.The geometric mean is useful for when quality scores don’t share the scale.
>
> > What if the value is negative?
>
> This was rare in our environments(seen early in swimmer). When a negative or zero valued Q exists in the tree, we modify the geometric mean calculation by adding |min(Q) + \delta| to the Q estimates. Where \delta is a positive small non-zero term which could be optimized (https://www.arpapress.com/volumes/vol11issue3/ijrras_11_3_08.pdf) to more closely recover the geometric mean.
>
> > Figure 1 looks the same as Figure 1 in https://arxiv.org/pdf/1909.07373.pdf, but I don't find any reference in the paper.
>
> Good catch, we will add the proper reference to fix this.
>
>  > How is PTN compared to model-based RL algorithms?
>
> We do, see Section 4.4. We use PPN(background planning) as a baseline and this is considered an implicit model-based method. To the best of our knowledge, PTN is the first work to perform decision-time planning with an implicit dynamics model. We didn’t compare to an explicit model-based(ex: observation prediction network) as this was shown to perform poorly in VPN.
>
> In the introduction we talk about two categories of model-based reinforcement learning, “Explicit dynamics models are when the observations are directly being reconstructed. The second is implicit dynamics models. Implicit Dynamics models are when the dynamics model is learned indirectly.” This is further expanded on in related works(ex: PPN)
>
> > More importantly, note that the policy in PPO is stochastic, so how is PTN compared to the deterministic policy?
>
> PTN is also stochastic (when b<\infty) so we don’t see a reason to compare with deterministic variants. It’s common(ex: ATreeC experiments) to compare with the stochastic variant(as opposed to a deterministic version).
>
> > How is the proposed pi-Q-backup method compared to classical control method, e.g. MPC?
>
> Simply put, MPC is when you replan at every step. PTN at evaluation time performs MPC.
>
> > Does the proposed planning algorithm work for model-free algorithms?
>
> The proposed planning algorithm(w/ depth=1) could work for model-free algorithms if you have access to a Q-function and a policy.
>
> > As this paper talks about planning with implicit dynamics models, how is the proposed method compared with explicit dynamics models?
>
> We did not test this as Observation Prediction Network(explicit model-based method) was shown to perform poorly in VPN.
>
> Minor comments:
> > In Algorithm 1 Line 11, could you please check the brackets?
>
> Thanks, we will fix that for the camera ready version
>
> > Page 4, "Thus, cumulative density function (cdf) of pi_F is given by ...": Could you please check the correctness of the equation?
>
> In the camera ready version we can update the range of the integral to be from -\infty to z
>
> > What does "worst-case" in Theorem 1 mean?
>
> Given any policy pi (learned by PTN), worst-case is the maximum possible difference between the mean of pi_F and pi over all possible choices of Q-functions.
>
> > How is correlation in Table 2 calculated?
>
> We calculate correlation from sample covariance divided by the product of sample standard deviations. In Section 4.2 we say, “To measure correlation we first train a PPN policy for 200 thousand time-steps on the MuJoCo environments. Next we collect trajectories over 10 thousand timesteps. Then at each observation we uniformly sample 100 actions and compute the corresponding Q-values and corresponding PDF points from $\pi$. This gives us 10 thousand estimates of correlation. From here we fit a normal distribution to the correlation results and report them in Table 2.”
>
> > In Algorithm 1, is the return value a scalar or a vector?
>
> A vector of Q/pi-Q values dependent on the branching factor. We will make this more clear in the camera ready version.
>
> > One can have infinite b but sample a uniformly to optimize Q (and then pi_F becomes maxQ policy), so I don't think b can be simply characterized as the confidence.
>
> The statement is for when sampling is done based on pi and b is finite. The second paragraph on page 4 states that “we sample b actions based on pi”. If sampling is not based on pi (e.g., uniform sampling), then pi_F does not depend on pi. Furthermore, if infinite b is used, sampling based on any distribution with non-zero pdf for all the actions will result in pi_F becoming maxQ policy, as mentioned in the second bullet point on page 3 of the paper.

---

> > ### Comment · AnonReviewer1 · 2019-11-14
> > **Initial response**
> >
> > I didn't go through all details in your response but I still have questions on my main concern, that is, I don't understand why sqrt(value * probability) makes sense.
> >
> > Suppose there is an MDP (with a fixed horizon or infinite horizon), with rewards in [-1, 1]. Now I subtract 100 from every reward, so the rewards are in the range of [-101, -99]. Note that the original optimal policy is still the optimal policy. However, all Q values are now negative. It might be possible that you didn't encounter negative Q values in your experiments, but I do think it's a very common case.
> >
> > Moreover, the given formula (sqrt(value * probability)) is not linear to the reward function, making it hard to interpret what this equation computes. It can't be any Q function, as Q function is linear to reward.

---

> > > ### Author Response · Authors · 2019-11-14
> > > **handling negatives**
> > >
> > > Thanks again for taking the time to share your thoughts. I've replied inline below
> > >
> > > > Suppose there is an MDP (with a fixed horizon or infinite horizon), with rewards in [-1, 1]. Now I subtract 100 from every reward, so the rewards are in the range of [-101, -99]. Note that the original optimal policy is still the optimal policy. However, all Q values are now negative. It might be possible that you didn't encounter negative Q values in your experiments, but I do think it's a very common case.
> > >
> > > You bring up valuable concerns. As we previously mentioned, “When a negative or zero valued Q exists in the tree, we modify the geometric mean calculation by adding |min(Q) + \delta| to the Q estimates. Where \delta is a positive small non-zero term which could be optimized (https://www.arpapress.com/volumes/vol11issue3/ijrras_11_3_08.pdf) to more closely recover the geometric mean.” Though we note that there is a typo, we actually add |min(Q)| + \delta when a negative or zero valued Q exists.
> > >
> > > Shifting the Q function results by a constant doesn’t change the argmax if both Q and \pi are optimal. Thus the optimal policy would not change.
> > >
> > > > Moreover, the given formula (sqrt(value * probability)) is not linear to the reward function, making it hard to interpret what this equation computes. It can't be any Q function, as Q function is linear to reward.
> > >
> > > Perhaps we are misleading with notation. We are not trying to say that the backup resembles the true Q function. The purpose of the backup is to choose an action that will lead to max expected returns. If the backup was exclusively using the Q approximations from the reward and value networks then it could be fair to say the backup resembles the Q function. However, we are arguing that both \pi and the Q composition(from the value and reward networks) are both signals of how “good” an action is to take. Thus the augmented geometric mean takes both quality scores into account. The argmax of \pi_F leads to larger expected returns than \pi or Q does individually and our experiments justify this.
> > >
> > > For the camera ready version we will change the notation in Section 3.2.2 to avoid misleading readers that the \pi-Q backup resembles the Q function.

---

### Official Review · AnonReviewer3 · 2019-10-23
**Official Blind Review #3**

**Rating:** 1

**Review:**

This paper claims to present a method for combining model based and model free approaches. The paper I find very poorly written hence my certainty about understanding the method cannot be very high. In training the method seems to build up a backup tree using transition operators and a policy and using them as targets for learning. In training it is not quite as clear what they are doing. The paper seems novel and sensible and has some experimental results that are not trivial but the writing is so difficult to follow that it makes it impossible for me to assess the contributions and even check correctness. I also think that readers would find it too difficult to understand as well. This is making a complete rewrite mandatory. I have added some initial pointers that would help making this more readable but implementing these would only allow us to assess what is being done rather than guarantee acceptance.

Since the rebuttal is not intended as a deadline extension I recommend rejecting this paper!

Major points:
* I find the related work quite badly written. There is content but what the reader cares about it situating the paper in the landscape of existing methods. There is none of that here: why do we care in this work about PPO and not say Q(\lambda). It should build up the components that were existing in the literature not just present some other methods. It needs to tell us roughly what is similar in this work to what was previously existing (roughly at least).
* If PPN is so central it has to be presented before PTN and notation should be introduced there. Introducing formulas without explaining notation like eq (1-4) serves only to alienate the reader and the (well-intentioned) reviewer.
* Please separately present how inference works and present the learning all in one place instead of losses in 3.1 and how to construct targets spread out until 3.2.2
* The fact that you need so many "note that " should be a red flag that the writing is not right (12 times).
* The algorithm was the most useful thing in the paper but even there it should be much clearer e.g. what is Q, how come we can write Q[j] = in consecutive lines. The second one should probably be Q[j] +=. I can't be sure because everything else is so hard to track.
* To me if you have a network that takes in previous step and action and produces a latent next step that is an explicit transition model. How is it not ?

**Experience Assessment:**

I have read many papers in this area.

**Review Assessment: Checking Correctness Of Derivations And Theory:**

I did not assess the derivations or theory.

**Review Assessment: Checking Correctness Of Experiments:**

I did not assess the experiments.

**Review Assessment: Thoroughness In Paper Reading:**

I read the paper at least twice and used my best judgement in assessing the paper.

---

> ### Author Response · Authors · 2019-11-13
> **Response to Reviewer #3**
>
> Thanks for taking the time to review and give feedback. We’ve reviewed your points and will respond in line below.
>
> > why do we care in this work about PPO and not say Q(\lambda)
>
> PTN focuses on leveraging an implicit dynamics model to perform decision-time planning in continuous action spaces. We specifically went over works related to policy gradient methods and continuous action space(PPO) and implicit dynamics models(ATreeC/TreeQN/VPN/PPN). While Watkins Q(\lambda) is interesting, it neither relates to operating in continuous action space nor building an implicit dynamics model. However, you bring up a fair point that PPO is not entirely necessary. In the camera ready version we will drop the PPO subsection and use the extra space to provide more details about PPN.
>
> > If PPN is so central it has to be presented before PTN and notation should be introduced there
>
> You bring up a good point, some explanations are currently in the appendix(ex: clipping and importance sampling ratios) but we can improve. In the camera ready version we will include a more in depth explanation of PPN in related works and additionally will provide extra details in the appendix(ex: PPN’s training algorithm).
>
> > Please separately present how inference works and present the learning all in one place instead of losses in 3.1 and how to construct targets spread out until 3.2.2
>
> Targets for 3.1(eq 2-4) include observed rewards, bootstrapped n-step returns, and generalized advantage estimates. Q and \pi-Q backup found in 3.2 are not used as targets in eq(2-4), they are only used for  decision-time planning at evaluation time.
>
> In the first paragraph section 3 we say:
> “Actions during behavior time are chosen by a model-free policy. Learning is done with a model-based approach that follows the behavior policy’s rollout trajectory.”
> “a latent space transition model is embedded into the architecture. The embedded latent space transition model allows us to backpropagate from multiple simulation steps into the future back to a grounded observation. As a consequence, a dynamics model is learned”
>
> This means that the behaviour policy does not use the model. During training the objectives force us to make use of dynamics model at any point i>0(equation 1). In the camera ready version we can expand on the implications of the above quotes to explicitly say how inference is done.
>
> In short:
> Behaviour time: no transition model and no decision-time planning
> Training time: transition model but no decision-time planning(because we follow behaviour trajectory)
> Evaluation time: transition model and decision-time planning planning
>
> > The algorithm was the most useful thing in the paper but even there it should be much clearer e.g. what is Q, how come we can write Q[j] = in consecutive lines. The second one should probably be Q[j] +=.
>
> It seems there is a misunderstanding, most likely stemming from a poor choice of notation on our part. The value set recursively at the first mention of Q[j] is used in the following line when applying td-\lambda. To be more clear we will replace the first mention with a temporary variable.
>
> > To me if you have a network that takes in previous step and action and produces a latent next step that is an explicit transition model. How is it not?
>
> We talk about this in the introduction, “Explicit dynamics models are when the observations are directly being reconstructed. The second is implicit dynamics models. Implicit Dynamics models are when the dynamics model is learned indirectly.” This is further expanded on in related works(ex: PPN)

---

> > ### Comment · AnonReviewer3 · 2019-11-15
> > **thanks for the clarifications i will have a read of the new draft**
> >
> > Thank you for the clarifications. I will have a look at the new draft and update my evaluation.
> >
> > I am reluctant, a priori, to change my opinion upon large rewrites because it promotes poorly written initial submissions and I think, as a community, we have quite enough of those. But I will give a fair assessment of the new draft.

---

### Official Review · AnonReviewer2 · 2019-10-23
**Official Blind Review #2**

**Rating:** 3

**Review:**

The paper proposes a modification to Policy Prediction Networks (PPN) in which the learned transition-, reward- and value function models are used at test-time in a planning procedure.
A second contribution is the "pi-Q-backup" which uses the geometric mean of both the policy and the value function as maximisation target of the planning step.

 Overall, I find the idea interesting and the experimental evaluations promising. However, I am voting for "weak reject" for the reasons outlined below. If some (or all) of them are address, I'd be happy to raise my score.

- I found the paper hard to understand. In particular, the algorithm PPN on which this work is build is not explained at all, requiring the reader to read the original PPN paper. Including a description of PPN, including it's main features, would greatly help the paper. Second, I am still not sure I correctly undestand when each component is used. As I currently understand it, the main usage of the model during training time is to compute the Advantages in equation (2)? Or are those computed based on rollouts? If so, where is the model actually being used during training?
- Figure 1 is taken directly from the PPN paper without any reference or citation (as far as I can tell).
- For the comparison in Figure 5, it would be great if PPN could also be tested with the newly introduced parameter \beta_i. At the moment, it is hard to tell whether the performance gains are due to \beta or due to the proposed planning scheme.
- I'm confused about Theorem 1: Wouldn't we want an upper bound on the difference of means?  Also, what does 'worst-case' mean? Is that for the 'worst' Q-function we could choose?

Edit:
Thank you for your comments and updated manuscript.

I think the writing has improved significantly, but could still be further improved and clarified. In particular, the question of how the model and other components at various points in time could be made more obvious. I found the authors' response to R3 here helpful as well.
At least for me some of the confusion arises not due to the complexity of the proposed approach, but just because combining real and 'simulated' transitions can be used and mixed in so many different ways that it's important to be clear about it. Also, at least personally, I found the explanation "Learning is done with a model-based approach that follows the behaviour policy’s rollout trajectory. However, the test policy follows a model-based approach" still not very helpful.
Overall, I think the presentation is on a good way but needs some more work.

With that being said and now having a better understanding of the algorithm I think this is very interesting work. However, I share R1's concerns about the computation of the \pi-Q backup, in particular that it seems arbitrary and doesn't handle negative values. I'm also not convinced that adding |min(Q)| is a good solutions as a) we don't always have access to that value and b) If I'm not wrong, than \pi_F is not invariant under a shift of Q.
I'm wondering why the authors decided to take the geometric mean instead of following the more typically used approach of using exp(Q/Temperature)*pi to combine Q-function and a policy distribution (see e.g. the "Control as Inference" literature, the "Maximum a Posteriori Policy Optimisation" or "Soft Actor Critic" algorithms, in particular the "Soft Value functions". I think this should at the very least be an ablation study, and could be even performing better and at the very least be robust to negative values.

Overall, I think this is very interesting work and could become a very strong paper, but I will remain to recommend a "weak reject" because I think it needs some more work to get there.

**Experience Assessment:**

I have read many papers in this area.

**Review Assessment: Checking Correctness Of Derivations And Theory:**

I assessed the sensibility of the derivations and theory.

**Review Assessment: Checking Correctness Of Experiments:**

I assessed the sensibility of the experiments.

**Review Assessment: Thoroughness In Paper Reading:**

I read the paper at least twice and used my best judgement in assessing the paper.

---

> ### Author Response · Authors · 2019-11-13
> **Response to Reviewer #2**
>
> Thanks for taking the time to review and give feedback. We’ve reviewed your points and will respond in line below.
>
> > Including a description of PPN, including its main features, would greatly help the paper.
>
> Good point. In the camera ready version we will include a more in depth explanation of PPN in related works. There currently exists details in the appendix regarding PPN’s approach to clipping but we will expand on this(ex: PPN’s training algorithm).
>
>  > the main usage of the model during training time is to compute the Advantages in equation (2)?
>
> Good question, this is our fault for not being explicitly clear. The model is not used to compute the advantages. In the camera ready version we will fix this. At training time the model helps with feature learning(via background planning). The main purpose of the implicit dynamics model is to be used at evaluation time to perform decision-time planning.
>
> > Or are those computed based on rollouts?
>
> Yes, when we expand the related work section on PPN we will make this more clear and reference that PTN follows the same approach to generating advantage estimates.
>
> > If so, where is the model actually being used during training?
>
> In the first paragraph of section 3 we say:
> “Actions during behavior time are chosen by a model-free policy. Learning is done with a model-based approach that follows the behavior policy’s rollout trajectory.”
> “a latent space transition model is embedded into the architecture. The embedded latent space transition model allows us to backpropagate from multiple simulation steps into the future back to a grounded observation. As a consequence, a dynamics model is learned”
>
> This means that the behaviour policy does not use the model. During training the objectives force us to make use of dynamics model at any point i>0(equation 1). We will make this more clear in the camera ready version.
>
> > Figure 1 is taken directly from the PPN paper without any reference or citation (as far as I can tell).
>
> Good catch, we will add the proper reference to fix this.
>
>  > For the comparison in Figure 5, it would be great if PPN could also be tested with the newly introduced parameter \beta_i. At the moment, it is hard to tell whether the performance gains are due to \beta or due to the proposed planning scheme.
>
> Fair point, we will include this comparison between PTN(no decision-time planning) and PPN in the appendix. An image of a preliminary table is linked(https://imgur.com/kIIrDNW), notice that these returns are worse than PTN(with decision-time planning) shown in Figure 5. This shows performance benefits from the modifications shown in Section 3.1(ex: \beta). The point of \beta is to stabilize returns over different values of training depth. In PPN the authors showed that optimal depth is highly dependent on the environment and returns can drastically differ. While the modifications certainly do not entirely fix this, we find that it mitigates large differences in returns from different training depths.
>
> > I'm confused about Theorem 1: Wouldn't we want an upper bound on the difference of means?
>
> As mentioned in the second paragraph of page 5, the goal of the theorem is to show that “The final policy pi_F can become significantly different from pi …. even when b = 2” (emphasis added). That is, the difference can be at least sigma/sqrt(pi), which is significantly large and not desirable. However, an upper bound would aim at showing the opposite. Another way to state the theorem is that there exist Q functions for which the difference between mean of pi_F and pi is at least sigma/sqrt(pi). Linked figure(https://imgur.com/FF4o2IS) shows that, as b increases, there exist Q functions for which pi_F and pi become increasingly different. We will add this to the appendix in the camera ready version.
>
> In the paper, we mention that “For smaller values of b, pi_F is more similar to pi and as b increases pi_F becomes similar to Q0.” That is, generally, because pi and Q are correlated (as shown in Table 2), we expect that on average, pi_F to be similar to pi (it may be interesting to study this theoretically, but we have not shown it). However, the theorem shows that there exists Q functions for which pi_F and pi are very different.
>
> > Also, what does 'worst-case' mean?
>
> Given any policy pi (learned by PTN), worst-case is the maximum possible difference between the mean of pi_F and pi over all possible choices of Q-functions.
>
> > Is that for the 'worst' Q-function we could choose?
>
> Yes, the worst-case is over all the possible Q-function.

---

> > ### Comment · AnonReviewer2 · 2019-11-13
> > **Initial quick response**
> >
> > Thank you for your response!
> > I haven't had the time to look at it in detail. However, upon a very first quick look you mention several updates you are planning on making to the camera ready version.
> >
> > In case you were not aware, I just wanted to mention that OpenReview allows updating the PDF already now, I believe until Friday (though I'm not sure).
> >
> > If you already have made changes to your manuscript, I'd encourage you to upload it already now.

---

> > > ### Author Response · Authors · 2019-11-15
> > > **Revisions**
> > >
> > > As per your and other reviewers suggestions/questions we have updated the draft. We stress that this is not final draft, but hope that it will clear up some questions.
> > >
> > > A brief summary of the updates:
> > > 1. drop PPO and give more details to PPN in related work
> > > 2. section 3.1 explicitly explain where targets come from
> > > 3. section 3.2 explain that decision-time planning has no impact on training
> > > 4.  appendix comparison of PTN(no decision-time planning) to PPN
> > > 5. add algorithm describing PPN to the appendix
> > > 6. footnote to explain how to handle negatives and zeros in the geometric mean calculation
> > > 7. missing PPN figure citation
> > > 8. appendix learning modifications
> > > 9. appendix numerical evaluation of theorem 1 bound
> > > 10. fix bracket in Algorithm 1
> > > 11. reduce ambiguity in Algorithm 1 by introducing a Q_tmp as a temporary variable
> > > 12. remove ambiguity on \pi_F integration bounds when showing the CDF
> > > 13. make clear that Algorithm 1 returns a vector

---

### Decision · Program_Chairs · 2019-12-19

**Decision:**

Reject

**Comment:**

The consensus amongst the reviewers is that the paper discusses an interesting idea and shows significant promise, but that the presentation of the initial submission was not of a publishable standard. While some of the issues were clarified during discussion, the reviewers agree that the paper lacks polish and is therefore not ready. While I think Reviewer #3 is overly strict in sticking to a 1, as it is the nature of ICLR to allow papers to be improved through the discussion, in the absence of any of the reviewers being ready to champion the paper, I cannot recommend acceptance. I however have no doubt that with further work on the presentation of what sounds like a potentially fascinating contribution to the field, the paper will stand a chance at acceptance at a future conference.